# Goal-Directed Planning and Goal Understanding by Extended Active Inference: Evaluation through Simulated and Physical Robot Experiments

**DOI:** 10.3390/e24040469

**Published:** 2022-03-28

**Authors:** Takazumi Matsumoto, Wataru Ohata, Fabien C. Y. Benureau, Jun Tani

**Affiliations:** Cognitive Neurorobotics Research Unit, Okinawa Institute of Science and Technology, Okinawa 904-0495, Japan; takazumi.matsumoto@oist.jp (T.M.); wataru.ohata@oist.jp (W.O.); fabien.benureau@oist.jp (F.C.Y.B.)

**Keywords:** active inference, teleology, goal-directed action planning

## Abstract

We show that goal-directed action planning and generation in a teleological framework can be formulated by extending the active inference framework. The proposed model, which is built on a variational recurrent neural network model, is characterized by three essential features. These are that (1) goals can be specified for both static sensory states, e.g., for goal images to be reached and dynamic processes, e.g., for moving around an object, (2) the model cannot only generate goal-directed action plans, but can also understand goals through sensory observation, and (3) the model generates future action plans for given goals based on the best estimate of the current state, inferred from past sensory observations. The proposed model is evaluated by conducting experiments on a simulated mobile agent as well as on a real humanoid robot performing object manipulation.

## 1. Introduction

In studying contemporary models of goal-directed actions of biological and artificial agents, it is worthwhile to consider these models from the perspective of a teleological framework. Teleology is a philosophical idea that originated in the days of Plato and Aristotle. It holds that phenomena appear not through their causes, but by their end results. While teleology is controversial and largely abandoned as a means of explaining physical phenomena, the idea has been extended to model action generation. A teleological account explains actions by specifying the state of affairs or the event toward which the actions are directed [1,2]. More simply, actions are generated to achieve purposes or goals. In psychology, Csibra et al. [3] propose that a goal-directed action can be explained by a well-formed teleological representation consisting of three elements: (1) a goal, (2) actions intended to achieve the goal, and (3) constraints, which are physical conditions that impose limits on possible actions. In a well-formed teleological representation of an action, the action should be seen as an effective means to achieve the goal within the constraint of reality. This account predicts that agents who represent goal-directed actions in this way should be able to infer unseen goals or unseen reality constraints, given the two remaining elements. Csibra et al. [3] verified this hypothesis by showing that even one-year-old infants can perform such inferences in their experiments.

Brain-inspired models for goal-directed action have been developed in various forms. Most brain-inspired models are based on forward models [4,5,6] for predicting sensory outcomes of an action to be executed. For goals given in terms of a preferred sensory state at the distal (terminal) step, optimal action sequences leading to the goal under assumed constraints, such as a minimal torque change criterion, can be obtained inversely using the forward model. Recently, Friston and colleagues [7,8] advanced the idea of goal-directed action using a forward model by incorporating a Bayesian perspective. Goal-directed planning for achieving the preferred sensory states is formulated under the framework of active inference [9,10,11] based on the free energy principle [12].

The aforementioned studies on goal-directed action can be expanded in various ways. One such possibility concerns the goal representation. Although goals in the aforementioned models are represented by a particular sensory state at each time step or at the distal step, some classes of on-going processes can also be goals for generating actions. For example, one can say “I got up early this morning to run”. In this case, the goal is not a particular destination, but the process of running. In the teleological framework, goals or the purpose of actions could be states, events, or processes which could be either specific or abstract and conceptual. What sorts of models could incorporate such diverse forms of goal representations? Another possibility for exploration is to incorporate the capability for inferring unseen goals or unseen reality constraints provided with the remaining two elements among actions, constraints and goals, as described by Csibra et al. [3].

For the purpose of examining these possibilities, the current study proposes a novel model by extending our prior goal-directed planning model, goal-directed latent variable inference (GLean) [13]. GLean was developed following the free energy principle [12] and it operates in the continuous domain using the predictive coding-inspired variational recurrent neural network (PV-RNN) [14]. The newly proposed model, GLean with teleology-inspired ideas (T-GLean), has three new key features. First, goals can be specified either by temporal processes (such as the earlier example of “going out for a run”) or static sensory states, for example, “going to the corner store”. Second, the model can infer goals by sensory observation, as well as by generating goal-directed action plans. This allows agents using the T-GLean model to anticipate goals and future actions by observing the actions of another agent. Third, the model generates future action plans for given goals based on the best estimate of the current hidden state, using the past sensory observations. Here, an action plan is a proprioceptive-exteroceptive sequence that achieves specific goals. Action plans are generated by mapping from the latent variables that are inferred by active inference. While the third feature is not necessarily novel, given that functions of sensory reflection or postdiction [15] have been examined using deterministic generative RNN models [16] and Bayesian probabilistic generative models [8], this feature is added to the model so that robots using T-GLean can generate goal-directed plans online while actions are being executed, whereas GLean can only generate plans offline.

Our proposed model is evaluated in Section 4 by conducting two experiments using two robot setups. The first experiment uses a simple simulated mobile robot that can navigate an environment with an obstacle, and the second experiment uses a physical humanoid robot with a larger degree of freedom that tracks and manipulates objects. In both experiments, the robots are trained to achieve two different types of goals. For the mobile robot, one type of goal is to reach a specified position, and the other type of goal is to move around a specific object repeatedly. For the humanoid robot, one type of goal is to grasp an object and then to place it at a specified position and the other type of goal is to lift an object up and down repeatedly. In both experiments, the trained robots are evaluated in generating goal-directed actions for specified goals, as well as to infer corresponding unseen goals for given sensory sequences. We also touch on generating action plans based on the rationality principle [3]. The following section describes related studies in more detail so that readers can understand easily how the proposed model has been developed by extending and modifying those prior proposed models.

## 2. Related Studies

First, we look in detail at how a goal-directed action can be generated using the forward model. Kawato et al. [4] proposed that multiple future steps of proprioception (joint angles) as well as the distal position in task coordinate space of an arm robot can be predicted, given the motor torque at each time step, by training the forward dynamics model and the kinematic model that are implemented in a feed-forward network cascaded in time. After training, optimal motor commands to achieve the desired positions at the distal step following the minimal torque change criterion can be computed using the prediction error information generated in the distal step. To deal with the hidden state problem encountered by real robots navigating with limited sensors, Tani [17] extended this model by implementing the forward dynamics model in a recurrent neural network (RNN) with deterministic latent variables. It was shown that a physical mobile robot can generate optimal, goal-directed navigation plans with the minimum travel distance criterion in a hidden state environment using that proposed model. Figure 1a depicts the scheme of goal-directed planning using a forward dynamics model implemented in an RNN.

In the graphical representation in Figure 1a, $a_{t}$, $d_{t}$ and ${\bar{x}}_{t}$ denote the action, deterministic latent variable, and sensory prediction at time step *t*, respectively. ${\hat{x}}_{T}$ is the sensory goal image at the distal step *T*. Within the scheme of the forward model and active inference, the policy is a sequence of actions that is optimized such that the error between the sensory goal image and the preferred image at *T* can be minimized (the minimization criterion such as the travel distance is omitted for brevity).

Friston and colleagues [7,8] formulated goal-directed actions by extending the framework of active inference [9,10,11], which is based on the free energy principle [12]. The major advantage of this approach, compared to the aforementioned conventional forward model, is that the model can cope with uncertainty that originates from the stochastic nature of the environment by incorporating a Bayesian framework. Originally, the free energy principle was developed as a theory for perception under the framework of predictive coding, wherein probabilistic distributions of past latent variables in generative models are inferred for the observed sensation by minimizing the free energy. Later, active inference was developed as a framework for action generation wherein action policies as well as the probabilistic distribution of future latent variables in generative models are inferred by minimizing the so-called expected free energy. Minimizing the expected free energy $G_{aif}$ for the future maximizes both the epistemic value shown in the first and the second terms, and extrinsic value in the third term in Equation (Equation 1).
(1)$$G_{aif}=\sum_{t>t_{c}}^{T}-E_{q(z_{t},x_{t}|\pi )}\left[\;\;\;\underset{\mathrm{epistemic}\;\mathrm{value}}{\underset{︸}{\log p\left(z_{t}|x_{t}\right)-\log q\left(z_{t}|\pi \right)}}\;\;+\underset{\mathrm{extrinsic}\;\mathrm{value}}{\underset{︸}{\log p(x_{t})}}\right]$$
where π is an optimal policy or a sequence of actions, $p(z_{t}|x_{t})$ in this case is the posterior distribution, $q(z_{t}|\pi )$ is the approximate posterior predictive distribution (which for brevity, we will refer to just as the posterior predictive distribution), $p(x_{t})$ is sensory evidence, and $t_{c}$ is the current time step. The extrinsic value represents how much the sensory outcomes expected in the generated future plan are consistent with the preferred outcomes. The epistemic value, on the other hand, represents the expected information gain with predicted outcomes. This means that this value indicates the expected decrease of uncertainty with regard to the hidden state provided by the sensory observation.

Augmentation of expected free energy has also been used to create interesting agent behavior, such as modulating the epistemic value to simulate top-down attention [18], or switching between exploratory and habituated behavior based on the environment [19]. In summary, an optimal action policy for achieving the preferred sensory outcomes and the reduction of uncertainty of hidden states can be generated by minimizing the expected free energy.

Figure 1b depicts this framework, wherein the probabilistic latent variables from timestep 0 to *T* (hereafter noted as 0:*T*) $z_{0:T}$, as well as an action policy, $a_{0:T}$ are inferred by minimizing the expected free energy. In order to find the optimal policy, a search can be run in policy space, such as a Monte-Carlo tree search [18]. The expected free energy at each future timestep can be computed by policy rollout, before moving the agent [20]. Friston and colleagues [7,8] implemented this framework mostly in discrete space, Hafner et al. [21], and proposed an analogous model implemented in continuous state problems.

Matsumoto and Tani [13] proposed another type of goal-directed action plan generation model, GLean, which is based on the free energy principle using a variational RNN. The main difference between this scheme and those proposed by Friston and colleagues [7,8] and by Hafner et al. [21] is that an optimal goal-directed plan is obtained by inferring the lower-dimensional probabilistic latent space, instead of the higher dimensional action space. This scheme is inherited from prior work [22] using a deterministic multi-layered RNN for hierarchical action planning, and it is also analogous to the recent development of sub-goal embedding in latent space using model-based reinforcement learning, as shown in [23]. The graphical model of the scheme is depicted in Figure 1c, wherein the proprioception ${\bar{x}}_{t}^{p}$ and the exteroception ${\bar{x}}_{t}^{e}$ at each time step *t* are predicted by the learned generative model using the probabilistic latent variable $z_{t}$ and deterministic latent variable $d_{t}$. The use of both deterministic *d* and probabilistic latent variables *z* is inherited from the original design of PV-RNN. The idea behind this design is that while a generative model that can generate complex perceptual sequences requires a latent state space with a certain number of dimensions, the essential degrees of fluctuation at each time step are limited. Thus, it is possible to have a smaller of number of probabilistic latent variables compared to the number of deterministic latent variables. For a given preferred goal represented as an exteroceptive state at the distal step *T* as ${\hat{x}}_{T}$, a proprioceptive-exteroceptive sequence expected to reach this goal state is searched by inferring an optimal sequence of the posterior predictive distribution of the latent variables $q(z_{1:T})$ by means of minimizing the expected free energy $G_{gl}$ as shown in Equation (Equation 2).
(2)$$G_{gl}=\underset{\mathrm{goal}\;\mathrm{error}}{\underset{︸}{-E_{q(z_{T}|{\hat{x}}_{T})}\left[\log p(x_{T}|d_{T})\right]}}+\underset{\mathrm{complexity}}{\underset{︸}{\left(\sum_{t=t_{c}}^{T}D_{KL}\left[q\left(z_{t}|{\hat{x}}_{T}\right)\left|\right|p\left(z_{t}|d_{t-1}\right)\right]\right)}}$$

Here, the expected free energy is represented by a sum of two terms, namely the goal error at the distal step, shown in the first term, and the complexity, summed over all future steps, shown in the second term. In our previous work [13] evidence was conditioned on *z* instead of *d*; however, since $d_{t}$ depends on $z_{1:t}$, the integration can still be applied.

Complexity is the divergence between the posterior predictive distribution and the prior distribution of the probabilistic latent variables, and is described in more detail later in Section 3. By minimizing this form of free energy, plans reaching the preferred goal states, but following well-habituated trajectories learned during learning, can be generated. We note that this model does not infer the policy directly. Instead, the motor command or action *a* at each time step is computed by means of the proprioceptive error feedback by sending the prediction of the next step proprioception as the target to the motor controller when the agent generates movement.

## 3. The Proposed Model

Our newly proposed model, T-GLean, is an extension of GLean [13] and also uses the PV-RNN architecture, which leverages the idea of multiple timescale RNN [13,24] with the expectation of development of a functional hierarchy through learning. In the proposed model, both proprioception and exteroception are necessary, since they are not always correlated. For example, proprioception and exteroception can be correlated when a robot is moving an object at which it is looking, but not when the robot is trying to reach an object that is static on a table. The proposed model also employs a novel goal representation scheme. Each goal is represented by its category, for, e.g., reaching or cycling, associated with optional parameters, for, e.g., position or speed. The network is designed to output the expected goal at each time step based on learning, as shown in Figure 2.

When the current preferred goal is given at each time step, the posterior predictive distribution is updated in the direction of minimizing the divergence between the preferred goal and the expected goal at each time step. This update of the posterior predictive distribution at each time step is conducted following the backpropagation through time (BPTT) scheme [25]. By taking mappings from the posterior predictive distribution through the deterministic latent variable, the expected proprioceptive and exteroceptive trajectory leading to the goal can be generated. This proprioceptive-exteroceptive trajectory represents a goal-directed plan. There are some notable differences between our proposed scheme for goal-directed plan generation and schemes based on a conventional active inference framework, such as in [8], where the expected free energy is computed for every possible policy by inferring the optimal posterior predictive distribution for minimizing the expected free energy. Then, the policy that can best minimize the expected free energy is selected as the optimal plan. In contrast, in our proposed model, optimization for minimizing the expected free energy is conducted in the space of the posterior predictive distribution by stochastic gradient descent (SGD), as described in our previous work [13]. Once the optimal approximate posterior distribution at each time step is determined, the corresponding proprioception and exteroception at each step is also determined by mapping through the deterministic latent variable. Here, the obtained expected proprioceptive sequence can be taken as an action policy, as the feedback controller that is attached to the output of the model can generate optimal motor control sequences to achieve the expected proprioceptive sequence.

The network is operated in the following three modes, each of which minimizes free energy as its loss function. In learning mode, predictive models are learned using provided training sequences by minimizing the evidence free energy, which is free energy where sensory evidence is available. In the online action generation mode, a future goal-directed plan is generated by minimizing the expected free energy while the past sensory experience is reflected by minimizing the evidence free energy. As this occurs in real-time, the past sensory experience is constantly updated online. In the goal inference mode, the expected goal is inferred using the observed exteroceptive sequence by minimizing the evidence free energy. For a given exteroceptive sequence, the posterior predictive distribution is updated in the direction of minimizing the divergence between the given exteroceptive sequence and its reconstructed exteroceptive sequence. This generates the expected goal in the outputs at each time step. The following sub-sections describe the model in more detail.

### 3.1. Model Architecture

Figure 3 depicts the employed architecture as a graphical model consisting of three layers of PV-RNN. It is similar to the architecture employed in [13], with the modification introduced in [26] that has a top-down connection from higher layers to lower layers $d_{l+1}\to d_{l}$ in the same time step, rather than from the previous time step. The graphical model shows the RNN unrolled into to the past as well as into the future, with the current time step at $t_{c}$. Each layer indexed by *l* contains the probabilistic latent variable *z*, from which $z_{l,t}^{p}$ is sampled from the prior distribution, and $z_{l,t}^{q}$ the approximate posterior distribution for the past and the posterior predictive distribution for the future, as well as the deterministic latent variable $d_{l,t}$, for each time step *t*. The probabilistic latent variables can capture probabilistic fluctuations, while the deterministic variables capture the average trajectory. Please note that deterministic variables take as input the deterministic variable from the layer above except on the top layer. The output of the bottom layer (L1) is split into proprioception ${\bar{x}}_{t}^{p}$, exteroception ${\bar{x}}_{t}^{e}$, expected goal ${\bar{g}}_{t}$, and distal step ${\bar{s}}_{t}$. Unless noted otherwise, the preferred goal $\hat{g}$ is given at all time steps as a target. Where available, the observed proprioception $x_{t}^{p}$ and exteroception $x_{t}^{e}$ are used as targets for error minimization. The distal step $s_{t}$ is a discrete random variable that takes the value 1 if *t* is the distal step of a sequence of steps that completes a given goal. During training, $\hat{s}$ is a one-hot vector in the time dimension with a single peak at the time step in which the goal is actually achieved. The distal step is not known during plan generation, but can be estimated by the trained network. The output $\left(\bar{x},\bar{g}\right)$ and targets $\left(x,\hat{g}\right)$ are softmax encoded; however, for brevity, the output layer that decodes the network output into raw output for the agent is not shown in this figure. In subsequent subsections, we will describe in more detail the aforementioned modes on which the network can operate. We do not make a complete derivation of PV-RNN in this paper, but focus on key aspects of this model.

### 3.2. Learning

Based on the graph connectivity shown in Figure 3, the forward computation of PV-RNN is given in Equation (Equation 3). Equation (Equation 3) is based on the formula of the continuous-time recurrent neural network (CTRNN) [27,28]. Multiple-timescale RNN (MTRNN) [24] uses multiple layers of CTRNN with different time constants assigned to each layer, building a hierarchical structure. PV-RNN further builds on MTRNN with the introduction of probabilistic latent variables, as described in Equations (Equation 4) and (Equation 5).
(3)$$\begin{array}{cc} h_{l,t} & =\left(1-\frac{1}{\tau^{l}}\right)h_{l,t-1}+\frac{1}{\tau^{l}}\left(W_{d,d}^{l,l}d_{l,t-1}+W_{z,d}^{l,l}z_{l,t}+W_{d,d}^{l+1,l}d_{l+1,t}\right),\\ d_{l,t} & =\tan h(h_{l,t}),\\ d_{l,0} & =0. \end{array}$$

$\tau^{l}$ is the time constant of layer *l*. The internal state of each RNN unit $h_{l,t}$ at time step *t* and level *l* is computed as a sum of the connectivity weight multiplication of $z_{l,t}$, $h_{l,t-1}$, and $h_{l+1,t}$ (if *l* is not the top layer). This is also shown graphically in Figure 3. The connectivity weight matrix W is indexed from layer to layer and from unit to unit. For brevity, bias terms have been omitted.

The probabilistic latent variable *z* follows a Gaussian distribution. Each sample of the prior distribution $z_{t}^{p}$ for layer *l* is computed as shown in in Equation (Equation 4).
(4)$$\begin{array}{cc} \mu_{l,t}^{p} & =\left\{\begin{array}{cc} 0, & \mathrm{if}\;t=1\\ \tan h(W_{d,z^{\mu^{p}}}^{l,l}d_{l,t-1}), & \mathrm{otherwise} \end{array}\right.\\ \sigma_{l,t}^{p} & =\left\{\begin{array}{cc} 1, & \mathrm{if}\;t=1\\ \exp(W_{d,z^{\sigma^{p}}}^{l,l}d_{l,t-1}), & \mathrm{otherwise} \end{array}\right.\\ z_{l,t}^{p} & =\mu_{l,t}^{p}+\sigma_{l,t}^{p}⊙\epsilon . \end{array}$$
where ϵ is a random noise sample such that ϵ∼N(0,I). Samples from $z_{t}^{q}$ as the approximate posterior distribution are computed as shown in Equation (Equation 5), where A is a variable to compute the mean and standard deviation for the approximate posterior at each step in a sequence. For brevity, here we assume we have a single sequence. If an output sequence is generated using the approximate posterior adapted during training, the corresponding training sequence should be regenerated.
(5)$$\begin{array}{cc} \mu_{l,t}^{q} & =\tan h(A_{l,t}^{\mu }),\\ \sigma_{l,t}^{q} & =\exp(A_{l,t}^{\sigma }),\\ z_{l,t}^{q} & =\mu_{l,t}^{q}+\sigma_{l,t}^{q}⊙\epsilon . \end{array}$$

To compute the output at time step *t*, there are three steps. First, the network output *o* is computed from $d_{1,t}$, as in Equation (Equation 6). This uses the output from layer 1 and treats the output layer as layer 0.
(6)$$o_{t}=W_{d,o}^{1,0}d_{1,t}.$$

We then compute the predicted probability distribution output $\bar{x}$. For this purpose, we use a softmax function to represent the probability distribution of the *i*-th dimension of the output as in Equation (Equation 7).
(7)$${\bar{x}}_{t}^{i,j}=\frac{\exp(o_{t}^{i,j})}{\sum_{j}\exp\left(o_{t}^{i,j}\right)}.$$
where ${\bar{x}}_{t}^{i,j}$ is the predicted probability that the *j*-th softmax element of the *i*-th output is on.

To explain the core part of the learning scheme following the free energy principle, we first describe how the model is formulated using the evidence free energy. For brevity, we assume there is a single layer only. This is shown graphically in Figure 4.

During learning, the evidence free energy shown in Equation (Equation 8) is minimized by iteratively updating the approximate posterior of *z*, as well as the RNN learning parameters *W* at each time step for all training sequences. The delta error between the generated output and the training output target, along with the Kullback–Leibler (KL) divergence between the approximate posterior and prior is backpropagated through time (BPTT) [25] from the end of the training sequence to the first timestep. The parameters of the network are updated in the direction of minimizing this delta error. For details on the use of BPTT in PV-RNN, please refer to [14].
(8)$$\begin{array}{cc} F(x,\hat{g},\hat{s},z)= & \sum_{t=1}^{T}(w\cdot \underset{\mathrm{complexity}}{\underset{︸}{D_{KL}\left[q\left(z_{t}|x_{t:T},{\hat{g}}_{t:T},{\hat{s}}_{t:T}\right)\left|\right|p\left(z_{t}|d_{t-1}\right)\right]}}\\ \; & -\underset{\mathrm{accuracy}}{\underset{︸}{E_{q(z_{t}|x_{t:T},{\hat{g}}_{t:T},{\hat{s}}_{t:T})}[\log p\left(x_{t},{\hat{g}}_{t},{\hat{s}}_{t}|d_{t}\right)}}]). \end{array}$$
where x, $\hat{g}$, and $\hat{s}$ are the observed sensory states, preferred goal, and distal step associated with the preferred goal, respectively. Free energy in this work is modified by inclusion of the meta-prior *w*, which weights the complexity term. *w* is a hyperparameter that affects the degree of regularization, similar to β in variational autoencoders [29]. We also note that since we are dealing with sequences of actions, the free energy is a summation over all time steps in the sequence.

The first term, complexity, is computed as the KL divergence between the approximate posterior and prior distributions. This can be expressed as follows:(9)$$D_{KL}\left[q\left(z_{t}|x_{t:T},{\hat{g}}_{t:T},{\hat{s}}_{t:T}\right)\left|\right|p\left(z_{t}|d_{t-1}\right)\right]=\int q\left(z_{t}|x_{t:T},{\hat{g}}_{t:T},{\hat{s}}_{t:T}\right)\log\frac{q(z_{t}|x_{t:T},{\hat{g}}_{t:T},{\hat{s}}_{t:T})}{p(z_{t}|d_{t-1})}dz_{t}$$

Given μ and σ for both prior *p* and posterior *q* distributions from Equations (Equation 4) and (Equation 5), respectively, samples from the distributions can be expressed as follows:(10)$$\begin{array}{cc} p(z_{t}|d_{t-1}) & =\frac{1}{\sqrt{2\pi {\left(\sigma_{t}^{p}\right)}^{2}}}\exp\left[-\frac{1}{2}{\left(\frac{z_{t}-\mu_{t}^{p}}{\sigma_{t}^{p}}\right)}^{2}\right],\\ q(z_{t}|x_{t:T},{\hat{g}}_{t:T},{\hat{s}}_{t:T}) & =\frac{1}{\sqrt{2\pi {\left(\sigma_{t}^{q}\right)}^{2}}}\exp\left[-\frac{1}{2}{\left(\frac{z_{t}-\mu_{t}^{q}}{\sigma_{t}^{q}}\right)}^{2}\right]. \end{array}$$

Thus, continuing from Equation (Equation 9), complexity can be computed as:(11)$$\int q\left(z_{t}|x_{t:T},{\hat{g}}_{t:T},{\hat{s}}_{t:T}\right)\log\frac{q(z_{t}|x_{t:T},{\hat{g}}_{t:T},{\hat{s}}_{t:T})}{p(z_{t}|d_{t-1})}dz_{t}=\log\frac{\sigma_{t}^{p}}{\sigma_{t}^{q}}+\frac{{\left(\mu_{t}^{q}-\mu_{t}^{p}\right)}^{2}+{\left(\sigma_{t}^{q}\right)}^{2}}{2{\left(\sigma_{t}^{p}\right)}^{2}}-\frac{1}{2}$$

For brevity, a case of a single *z*-unit consisting of a (μ,σ) pair is shown here. In practice, we can have multiple *z*-units, each with independent (μ,σ), and in that case, the RHS is a summation over all (μ, σ), as will be shown later in the experimental section. To calculate the second term of Equation (Equation 8), accuracy, we employ the same softmax encoding technique as in [30] wherein the target is converted into a softmax distribution. Following this, the minimization of KL divergence between the softmax encoded output and the softmax encoded target can be taken as equivalent to minimization of negative log-likelihood. During learning, Equation (Equation 8) is used as the loss function with the Adam optimizer, with the parameters noted Section 4.

### 3.3. Online Goal-Directed Action Plan Generation

A key difference between our newly proposed model, T-GLean, and our previous model, GLean, is the idea that the goal expectation ${\bar{g}}_{t}$ is generated at every time step instead of expecting the goal sensory state at the distal time step. Intuitively, this means that at every time step, the agent expects a goal state that the on-going action sequence will achieve. An advantage of this model scheme is that goals can be represented not only by distal sensory states to be achieved, but also by continuously changing sensory sequences. Another key feature of T-GLean is that the model generates plans online, in real-time, while the agent is acting on the environment, whereas our prior study using GLean showed only an offline plan generation scheme. In T-GLean the network maintains the observed sensory sequence in a past window while allocating a future window for the future time steps. In the past window, evidence free energy is minimized online by updating the posterior at each time step in order to situate all the latent variables to the observed sensory sequence. In the future window, the error between the preferred goal and the expected goal output is minimized for all steps by updating the posterior predictive distribution in the window iteratively, of which computation can be performed by minimizing the expected free energy. The future plan for achieving the preferred goals can be generated once the evidence free energy in the past window is minimized, i.e., the latent variables are well situated to the past sensory observation. This scheme is referred to as online error regression [16] and analogous models can be seen also in [8,31]. The scheme is shown graphically in Figure 5.

Generation going forward in time and error regression going backward are performed within a planning window of length win. Within the planning window, there is the past window of length $win^{p}$ and the future window of length $win^{f}$. In the current implementation, the length of the planning window is fixed, while the past window is allowed to grow up to half the length of the planning window. The current time step $t_{c}$ is one step ahead of the end of the past window. At the next sensorimotor time step, sensory information at $t_{c}$ becomes part of the past window, and $t_{c}$ moves forward one step, shrinking the future window. Once the past window is filled, the entire planning window slides one step to the right, discarding the oldest entry in the past window. During online planning, the network minimizes the plan free energy $F_{\mathrm{plan}}$ by optimizing the approximate posterior distribution within the past window and posterior predictive distribution within the future window. The goal error is backpropagated from the end of the planning window at $t_{c}+win^{f}$, while the sensory error is backpropagated from $t_{c}-1$ until the beginning of the planning window at $t_{c}-win^{p}$. This delta error is propagated to $z^{q}$, which is then updated in the direction of minimizing the delta error and the KL divergence between approximate posterior and prior. The forward computation and the error regression are repeated for a fixed number of iterations, such that a plan can be generated in a timely manner.

The plan free energy consists of the sum of the evidence free energy $F_{e}$ within the past window and the expected free energy *G* within the future window. This is expressed in Equation (Equation 12).
(12)$$\begin{array}{cc} F_{e}\left(x,\hat{g},z\right) & =\sum_{t=t_{c}-win^{p}}^{t=t_{c}}\left(w\cdot D_{KL}\left[q\left(z_{t}|x_{t:t_{c}},{\hat{g}}_{t:t_{c}}\right)\left|\right|p\left(z_{t}|d_{t-1}\right)\right]-E_{q(z_{t}|x_{t:t_{c}},{\hat{g}}_{t:t_{c}})}\left[\log p(x_{t},{\hat{g}}_{t}|d_{t})\right]\right),\\ G(\hat{g},z) & =\sum_{t=t_{c}}^{t=t_{c}+win^{f}}\left(w\cdot D_{KL}\left[q\left(z_{t}|{\hat{g}}_{t:t_{c}+win^{f}}\right)\left|\right|p\left(z_{t}|d_{t-1}\right)\right]-E_{q(z_{t}|{\hat{g}}_{t:t_{c}+win^{f}})}\left[\log p({\hat{g}}_{t}|d_{t})\right]\right),\\ F_{\mathrm{plan}} & =F_{e}+G. \end{array}$$

The expected free energy *G* shown in Equation (Equation 1), which follows [8], and our formulation of expected free energy in Equation (Equation 12) are quite different. This difference stems from the fact that the former is formulated for a given policy, whereas no policy is provided in the latter since the proprioceptive sequence is automatically determined from the optimized posterior predictive distribution, as described previously.

### 3.4. Goal Inference

Finally, before closing the current model section, we describe how future goals can be inferred from observed sensory sequences. Figure 6 shows a graphical model accounting for the mechanism of the goal inference with observation of the exteroception, but without actually generating actions. By observing the exteroception sequence from time step $t_{c}-win^{p}$ to the current time step $t_{c}$, the posterior $z^{q}$ at each step in the past window is optimized for minimizing the error between the observed and reconstructed exteroceptions. This results in an updated inference of the expected goal ${\bar{g}}_{t}$ for every time step *t*, with reduced error in the past and improved prediction for the future. In this way, the hidden state or intention of the action demonstrator can be inferred through exteroceptive observation. The inferred hidden state at each step can also be mapped to the expected goal at that step since such an association has been learned previously.

The network predicts simultaneously both the exteroception and proprioception for future steps leading to the inferred goal. We note that this scheme could be applied to the problem of inferring goals of other agents through observation of their movements, provided that the coordinate transformation between the allocentric view and the egocentric view can be made. For simplicity, the current study does not delve into this view coordinate transformation problem.

Equation (Equation 13) shows the modified evidence free energy used for goal inference, where only the observation of exteroception is used.
(13)$$F_{g}\left(x^{e},z\right)=\sum_{t=1}^{t=t_{c}}\left(w\cdot D_{KL}\left[q\left(z_{t}|x_{t:t_{c}}^{e}\right)\left|\right|p\left(z_{t}|d_{t-1}\right)\right]-E_{q(z_{t}|x_{t:t_{c}}^{e})}\left[\log p(x_{t}^{e}|d_{t})\right]\right).$$

## 4. Experiments

In order to test our proposed model, we conducted two experiments, one using a simulated agent and the other using a physical robot. In Experiment 1 (Section 4.1), we used a simulated mobile agent in 2D space in order to examine the model’s capacity in generating goal-directed actions and in understanding goals from sensory observation. Experiment 2 (Section 4.2) was carried out to test the model’s scalability in the real-world setting using a humanoid robot with higher degrees of freedom. T-GLean is implemented using LibPvrnn, a custom C++ library implementing PV-RNN that is currently under development. It is designed to be lightweight and to operate in real-time so interaction between agent and experimenter is possible. A pre-release version is made available under an open source license with instructions on reproducing the simulated agent experiments in the supplementary material.

### 4.1. Experiment 1: Simulated Mobile Agent in a 2D Space

We conducted a set of experiments using a simulated mobile agent that can generate goal-directed action plans based on supervised learning of sensorimotor experiences in order to evaluate the performance of T-GLean in the following four respects.

Generalization in learning for goal-directed plan generation;Goal-directed plan generation for different types of goals;Goal understanding from sensory observation for different types of goals;Rational plan generation.

Following our previous work [13], we first evaluate the generalization capability of the proposed model for reaching untrained goal positions using a limited number of teaching trajectories. The second test examines how the model can generate goal-directed plans and execute them for different types of goals, in this case reaching specified goal positions and cycling around an obstacle. The third test demonstrates the capability of the model to infer different types of goals from observed sensation (exteroception). The fourth test examines the model’s capability to generate optimal travel plans to reach specified goals under the constraint of the minimal travel time.

In each test, the simulated agent is in a square workspace with (x,y) coordinates in the range of [0.0,1.0] for both *x* and *y*. The agent always starts at position (0.5,0.1). In the center of the workspace is a fixed obstacle of size (0.3,0.05). The agent does not directly sense the workspace coordinates. Instead it observes a relative bearing and distance $\left(\theta_{t},\delta_{t}\right)$ to a fixed reference point at (0.0,0.5) as exteroception. In this experiment, the agent only receives exteroception without proprioception. The simulated agent controller handles conversion to and from the workspace coordinates to relative bearing-distance as the robot moves. This is computed as $\theta_{t}=atan2\left(y_{t}-0.5,x_{t}\right)$ and $\delta_{t}=\sqrt{x_{t}^{2}+{\left(y_{t}-0.5\right)}^{2}}$. Note that the agent is not trained to sense or avoid the obstacle. Rather the trained agent attempts to follow its trained paths around the obstacle as habituated actions.

At the onset of each test trial, the experimenter sets the preferred goal as a vector $\left({\hat{g}}_{t}^{\alpha },{\hat{g}}_{t}^{\beta }\right)$. ${\hat{g}}_{t}^{\beta }$ is a three dimensional one-hot vector representing the goal category, with each bucket representing reaching, clockwise cycling, and counter-clockwise cycling goals, respectively. ${\hat{g}}_{t}^{\alpha }$ is set as the goal position *x* coordinate if the reaching goal is set. Otherwise it is left as 0. We consider the distal step $s_{t}$ as a discrete random variable which takes the value 1 if *t* is the distal step of completing a given goal. The network estimates the probability of each time step being the distal step, and the agent stops at the time step with the highest estimated distal step probability, provided that it exceeds a predefined threshold value.

Unless stated otherwise, the experiments have different training data and separately trained networks; however, network parameters are identical between networks. Parameters used for each layer of the RNN are as shown in Table 1. Each network was trained for 100,000 epochs, using the Adam optimizer with parameters α=0.001, $\beta_{1}=0.9$, $\beta_{2}=0.999$. During planning, the parameters are slightly modified to α=0.04, 500 iterations per sensorimotor time step, and a planning window length of 70. The meta-prior *w* remains the same in all cases.

#### 4.1.1. Experiment 1A: Generalization in Plan Generation by Learning

From the perspective of data efficiency, it is important to examine how much the network can generalize unlearned parameters of goals, e.g., unlearned goal positions, from a limited number of learned experiences. In order to evaluate such goal generalization capabilities of the network in terms of goal position, we prepared four teaching datasets with decreasing numbers of goal locations to be reached, as shown in Figure 7. The goal locations are on a line in the range x=[0.2,0.8], y=0.9. The agent accelerates up to speed to a branching point, and then either turns left or right to the side of the obstacle, before moving toward the goal position. As the agent approaches the goal position, it decelerates to a stop.

The maximum trajectory length is 70, although the distal step occurs at around 40 time steps. To account for randomness in training, five networks with different initial random seeds are trained for each of the four datasets, for 20 trained networks in total. Untrained test goals were drawn from a uniformly random distribution in the range [0.2,0.8]. Each network was tested with ten untrained goals, with the results averaged over all test goals and networks for each dataset.

To evaluate goal position generalization, we considered the difference between the final agent position reached at the end of plan execution and the preferred goal at the end for each test trial, as well as the plan free energy that remained. The difference between agent position and goal is expressed as the root-mean-square deviation, normalized to the range of ${\bar{g}}^{\alpha }$ (NRMSD). We also consider the plan free energy at the point where the agent reaches the final position, following Equation (Equation 12). Figure 8 plots goal NRMSD and the mean plan free energy (plan FE) as the number of trained goals is changed. Plan FE is obtained by averaging $F_{\mathrm{plan}}$ over the length of the planning window.

We observed that the network achieved stable goal position generalization when at least 7 training trajectories are used. Furthermore, it can be seen that the plan free energy was minimized in a similar manner.

#### 4.1.2. Experiment 1B: Goal-Directed Plan Generation for Different Types of Goals

For this test, we prepared a more complex set of teaching trajectories, containing three distinct goals with significantly different patterns. The first goal, shown in Figure 9a, is similar to the previous goal-reaching trajectories shown in Figure 7a; however, this scenario is ill-posed, i.e., the teaching trajectories alternate between short and long paths for the same goal position. This set of teaching trajectories will also be reused in Experiment 1C and Experiment 1D. While the training data themselves are not ill-posed (i.e., a unique teaching trajectory exists for each teaching goal), the learning outcome is ill-posed due to position generalization, i.e., two different trajectories could be generated for an arbitrary specified goal position. We will revisit this issue in Section 4.1.4. For the second and the third goals, we prepared two training trajectories shown in Figure 9b, involve the agent cycling around the central obstacle in a clockwise direction and a counter-clockwise direction, respectively. In preparing teaching trajectories for the reaching goal, the time step of reaching a goal is set as the distal step and the remaining steps after this step are padded with the same value as those in the distal step, i.e., the agent remains stationary. Unlike the reaching goal, the cycling goals have no distal step. The training sequence demonstrates a single cycle without indicating the distal step, although the agent is expected to repeat cycling indefinitely in the action generation phase. There are 27 teaching trajectories (25 for the reaching goal trajectories, 2 for the two different cycling goal trajectories) in this training dataset, each of which has a length of 70 time steps. Training using these teaching trajectories was conducted under the learning conditions described previously.

After the training phase, we evaluated how precisely the agent can generate movement trajectories for achieving specified goals. To conduct plan generation, the experimenter sets a preferred goal $\hat{g}$. A variables as shown in Equation (Equation 5) are reset to zero for all steps. At each sensorimotor time step, the agent reads the current exteroceptive state from its sensors and the A variables are optimized for 500 iterations following the method in Section 3.3. After 500 iterations of optimization have been completed, the generated proprioceptive-exteroceptive sequence is saved as an action plan of the current time step. The controller takes the next anticipated proprioception state to generate the optimal motor commands for the agent to reach that state by the next sensorimotor time step. This process repeats until the agent reaches the estimated distal step or reaches a time limit for the experiment.

To evaluate the quality of goal-directed plan generation and its execution for these three different types of goals, we measured how closely the agent can reach the preferred reaching goals and also how closely it can generate cyclic movement trajectories compared to the corresponding training sequences. For this, we once again used NRMSD as we did in the previous experiment. In Table 2, we summarize the results for the reaching and cyclic goals. For the reaching goal, average NRMSD is given for 10 random untrained goal positions. For the cyclic goal, NRMSD between the agent’s movement and the training sequence for the entire 70 time step sequence is taken, and averaged for five clockwise and counter-clockwise orbits each. In the latter case, normalization is done over the entire dataset range rather than the goal range. Table 2 confirms that both types of goals can be achieved within minimal range of goal error.

Figure 10 shows three examples of plans generated and then executed for three goal categories. The visualization is split into three panels, with the left panel showing the planned and actual agent trajectories, the middle panel showing the sequence of proprioceptive-exteroceptive states in the planned trajectory at the top, while the bottom shows the history of sensory inputs and preferred goals. Finally the right panel shows mean evidence free energy for the past window and the mean expected free energy in the future window at the top, while *z* information is shown in the bottom. *z* information represents the effectiveness of the probabilistic latent variables in each layer. This value is computed by $D_{KL}\left[q||N\right]$ where *N* is the unit normal distribution, which considers that the approximate posterior probabilistic distribution with the unit normal distribution should have no effect, other than potentially as a source of random noise.

We observed that plan generation stabilized quickly after the onset of travel by minimizing the expected free energy, and the network was able to accurately estimate the distal step in the case of reaching the goal. Although the future plan trajectory was constantly changing due to the scheme of online plan generation and the error regression in the past, the motor plan generated was stably executed at each sensorimotor time step. It can be also observed that the evidence free energy in the past window continued to converge in all three plots, meaning that all latent variables were gradually situated to the behavioral context. Therefore, the deviation of the executed trajectory from the planned trajectory was limited. The full observed temporal processes can be seen in the recorded videos of the simulations at the following link (https://youtu.be/_aCMzSQsRi0 (accessed on 23 March 2022)).

#### 4.1.3. Experiment 1C: Goal Inference by Sensory Observation

Next, we evaluated how precisely the network trained in Experiment 1B can infer goals as well as future movement trajectories by observing movement trajectories in terms of the exteroception sequence. In this experiment, four movement trajectories achieving different goals are prepared, which reach goals located left and right of the goal line, clockwise cycling, and counter-clockwise cycling. The four test trajectories, shown in Figure 11, were generated in the same way as the training data, but with untrained goal positions for the reaching goals and with a slight variation and extended length for the cycling goals.

Figure 12 shows the anticipated trajectories and goals at different points in time as the network observed the goal-reaching trajectories in Figure 11a,b.

Initially, at t=1 (Figure 12a,d), before the agent observes any movement, the network makes a guess based on the learned prior distribution. Since the goal-reaching trajectories are most frequent in the teaching trajectory distribution, reaching a goal is inferred as a default goal when observing the movement trajectory at the starting point. Several steps later, at around t=8 (Figure 12b,e), the movement trajectory branches either left or right around the obstacle. We observed that in the case of the goal located on the left, the network initially anticipates a longer path going around the right side of the obstacle before observing that the movement trajectory goes around the other side of the obstacle. The observed phenomenon is due to the fact that the teaching trajectories contain two possible paths reaching the same goal position. Therefore, the network can generate two possible movement trajectory plans in the current ill-posed setting (This issue will be revisited in Section 4.1.4). As the movement trajectory approaches the goal area, the anticipated goal position ${\bar{g}}^{\alpha }$ is refined. By t=32 (Figure 12c,f), the goal is fully anticipated.

Figure 13 shows the trajectories and goals inferred for the observed cyclic trajectories shown in Figure 11c,d.

Until t=22 the observed trajectories are mostly indistinguishable from the reaching trajectories, of which the occurrence probability was learned as high in the prior distribution, the network infers goal-reaching as the default goal from these observations. While the observed movement trajectory begins to enter a cyclic trajectory after t=22, it still takes some time for the network to correctly infer the ongoing goal as cyclic. During this time, free energy increases, an example of which is shown in Figure 14. This is analogous to a ‘surprising’ observation. After some time, the goals are inferred correctly as the cycling goals as shown in Figure 13c,f for the counter-clockwise and clockwise cases, respectively. The free energy is reduced accordingly at this moment, as also shown in Figure 14, which shows a generated plan, the plan free energy, and *z* information in the counter-clockwise case. We note that the the effectiveness of the probabilistic latent variables (*z* information) also rises and stays elevated to contribute to producing the correct inference of possible goals.

We assume that the agent can recognize consecutive switching of goals from observation; thus, we also tested a scenario wherein the agent first observes a clockwise cycling trajectory, then a counter-clockwise trajectory, and a goal-reaching trajectory, all in one continuous sequence. This is a challenging test, since the network was not trained to cope with such dynamic goal switching. A video of this experimental result is provided at the following link (https://youtu.be/f-NixLAc48s (accessed on 23 March 2022)). In the animation it can be seen that the network inferred the changing goals successfully. However, the anticipated future trajectories were quite unstable toward the end of the trial. As we observed that free energy becomes quite large, it is presumed that the goal inference from the observation may take a relatively long time for convergence of the free energy for unlearned situations. Therefore, it may be difficult for the network to catch up to the goal switching if it occurs too frequently.

#### 4.1.4. Experiment 1D: Goal-Directed Planning Enforcing the Well-Posed Condition

In Experiment 1B, the network was trained using teaching trajectories that included alternative trajectories reaching similar goal positions as shown in Figure 9a. This made the goal-directed planning ill-posed since the network cannot determine an optimal plan between two possible choices under the current definition of the expected free energy. Figure 15 shows an illustrative example as the result of the ill-posed goal-directed planning where we see that both a short and long path can be generated for the same goal.

Conventionally, it has been shown that the problem of ill-posed, goal-directed planning can be transformed into a well-posed one by adding adequate constraints, including joint torque minimization [4] and travel distance minimization [17]. The current experiment shows an examination of the case using the travel time minimization constraint by adding an additional cost term in the plan free energy $F_{\mathrm{plan}}$ shown previously in Equation (Equation 12). The modified plan free energy $F_{\mathrm{plan}}^{\prime }$ is shown in Equation (Equation 14).
(14)$$\begin{array}{cc} \gamma_{t}= & tP(s_{t}|d_{t}),\\ F_{e}^{\prime }\left(x,\hat{g},z\right)= & \sum_{t=t_{c}-win^{p}}^{t=t_{c}}(w\cdot D_{KL}\left[q\left(z_{t}|x_{t:t_{c}},{\hat{g}}_{t:t_{c}}\right)\left|\right|p\left(z_{t}|d_{t-1}\right)\right]\\ & -E_{q(z_{t}|x_{t:t_{c}},{\hat{g}}_{t:t_{c}})}\left[\log p\left(x_{t},{\hat{g}}_{t}|d_{t}\right)-k\gamma_{t}\right]),\\ G^{\prime }\left(\hat{g},z\right)= & \sum_{t=t_{c}}^{t=t_{c}+win^{f}}(w\cdot D_{KL}\left[q\left(z_{t}|{\hat{g}}_{t:t_{c}+win^{f}}\right)\left|\right|p\left(z_{t}|d_{t-1}\right)\right]\\ & -E_{q(z_{t}|{\hat{g}}_{t:t_{c}+win^{f}})}\left[\log p\left({\hat{g}}_{t}|d_{t}\right)-k\gamma_{t}\right]),\\ F_{\mathrm{plan}}^{\prime }= & F_{e}^{\prime }+G^{\prime }. \end{array}$$
where γ is the added cost term for minimizing the travel time. This cost can be expressed by the summation of the estimated probability of becoming the distal step at *t*, given as $P(s_{t}|d_{t})$, multiplied by the time step length at each time step over all time steps in the planning window. For this experiment, we set the weight of the travel time cost k=0.1.

To evaluate the effect of adding the constraint for travel time minimization, we prepared three separately trained networks and four untrained goal positions that are shown with numbers overlaid on the training trajectories in Figure 16. The goal positions are selected to avoid the edges (lack of training trajectories) and the center (no difference in trajectory length).

The generated action plans for these test goal positions were classified as ‘short’ if the shorter of the possible paths (41 or fewer steps) was generated. Plan generation was repeated for each test goal position with 1000 different samples. We then calculated the probability of shorter plans being generated as $\sum_{\mathrm{samples}=1}^{1000}\left[{arg max}_{t}P\left(s_{t}|d_{t}\right)\leq 41\right]/1000$. The resultant probabilities for generating the shorter plans for each goal position with and without travel time cost are shown in Figure 17.

Without introducing the travel time cost γ, the probability of generating the short plans was around 50%, which is consistent with the ratio in the teaching trajectory set. This probability increased to over 90% with the addition of travel time cost to the plan free energy. These results confirm that the current modified model supports goal-directed plan generation in a well-posed manner.

### 4.2. Experiment 2: Object Manipulation by a Physical Humanoid Robot

To verify the performance of the proposed model in a complex physical world, we conducted experiments involving object manipulation by a humanoid robot, Torobo, manufactured by Tokyo Robotics Inc. Torobo was placed in front of an object manipulation workspace where a red cylindrical object was located for manipulation. Two types of goal-directed actions were considered. One was to grasp the object located at an arbitrary position in the workspace (36 cm × 31 cm) with both arms and then place it at a specified goal position on the goal platform (42 cm wide) fixed at one end of the workspace. The other type of goal was to grasp the object located at an arbitrary position in the workspace and to swing it up and down. The Torobo humanoid robot, red cylinder object, workspace, and goal platform are shown in Figure 18.

The neural network controlled Torobo’s two arms (6 degrees of freedom for each arm) and hip joints (2 degrees of freedom) to perform these goal-directed actions (total of 14 joint angles). The reading of these joint angles represents the proprioception of Torobo. Torobo can track the position of the red cylinder located in the workspace using a camera mounted in its head. The object position is constantly tracked by controlling the pitch and yaw of the camera head to keep the target object centered in the camera’s field of view. The red cylinder is visually located using YOLOv3 [32]. Therefore, the pitch and yaw of the head indicates the position of the object, and are considered to represent the exteroception of Torobo. Thus, exteroception can be represented by only a two-dimensional vector instead of a high-dimensional camera image. This simplification in visual image processing was necessary in order to generate goal-directed actions in real-time.

We conducted two experiments with Torobo. In Experiment 2A, we evaluated the performance in generating goal-directed planning and its execution in a similar fashion to Experiment 1B in Section 4.1.2, and in Experiment 2B we evaluated the capability of the network for goal inference by observation. The parameters used for each layer of the RNN are as shown in Table 3. As in Experiment 1, the network was trained for 100,000 epochs, using the Adam optimizer with a learning rate α=0.001, $\beta_{1}=0.9$, $\beta_{2}=0.999$. In order to maintain real-time operation with the robot, planning uses different parameters of α=0.1 and 100 error regression iterations per sensorimotor time step.

The training dataset consists of 200 trajectories in total, with 120 trajectories for the grasping and placing action, and 80 trajectories for the grasping and swinging action. To generate the training trajectories, source trajectories were first generated by interpolating joint angles between pre-recorded poses. Several poses for grasping, lifting, and placing were recorded by physically moving the robot by hand, with the remaining poses generated using inverse kinematics. To ensure the validity of the generated trajectories, they were played back on the robot and the resulting recorded joint trajectories used as the teaching samples.

The object is located at a random starting position within the workspace for each sample of the teaching trajectories. The goal position for placing is also randomly selected along the goal platform’s width. At each sensorimotor time step, we recorded 12 joint angles for both arms and two joint angles for the hip joints representing the proprioception and two head joint angles representing the exteroception along with a 3D vector representing the preferred goal and 1D scalar for the distal step marker. The maximum trajectory length was 204 time steps. The preferred goal $\left({\hat{g}}^{\alpha },{\hat{g}}^{\beta }\right)$ is represented in a similar manner to Experiment 1 wherein ${\hat{g}}^{\beta }$ is a 2D one-hot vector representing either goal of grasping-placing or grasping-swinging, and ${\hat{g}}^{\alpha }$ is a scalar representing the preferred goal position in the width direction of the goal platform in the case of the grasping-placing goal.

#### 4.2.1. Experiment 2A: Goal-Directed Plan Generation and Execution

To evaluate the performance of goal-directed plan generation and execution with the physical robot, we measured the RMS deviation to the ground truth, normalized to the data range, as shown in Experiment 1B. To examine the performance for achieving the grasping and placing goal, the experimenter placed the object at an arbitrary goal position on the goal platform, allowing Torobo to recognize the goal position by visual tracking. This position on the goal platform was measured and recorded as the ground truth. The object was then placed at a random position in the workspace, and the network started to generate action plans to achieve this specified goal while Torobo executed the generated motor plan simultaneously, in real-time. The difference between the final position of the object placed by the robot and the ground truth was then measured on the goal platform for 10 random goal positions. In the case of examining the grasping and swinging goal, the object was placed in three different positions in the workspace, and then the resulting robot trajectory in the swinging phase was compared to the closest teaching trajectory in the swinging phase. The result is shown in Table 4.

Compared to the results obtained in Section 4.1.2, the network generated a similar low deviation for achieving the grasping and swinging goal, while the deviation was higher for the grasping and placing goal. This is likely due to the relatively low precision in tracking the object, especially when placing the object on the goal platform, which was located at the far edge of the workspace. Figure 19 presents two plots showing an example of the plans generated when given the two types of goals. In a manner similar to that in Figure 10, the top panel shows the trajectory history of each joint angle of the robot. The second panel shows the planned joint trajectories. The third panel shows *z* information, and the bottom panel shows mean evidence and expected free energy. Example videos showing these experimental results can be seen at the following link (https://youtu.be/SZahOgFssC0 (accessed on 23 March 2022)).

#### 4.2.2. Experiment 2B: Goal Understanding

Finally, the experimental results for goal understanding are briefly described. In this experiment, the object was moved by the experimenter emulating the manipulation of the object by Torobo for each goal category. Torobo observed this object movement using object tracking while Torobo remained in the initial posture, except for its head joints, used for object tracking. The network in Torobo inferred the expected goal and predicted the future movement trajectory in terms of the sensory sequence, as shown in Experiment 1B. This experiment was repeated five times for each of the goal-directed grasping actions, placing and swinging. The network was judged to have correctly inferred the goal if the anticipated goal stably matched the experimenter’s actions, before plan execution or the experimenter’s actions ended. A video showing this experiment can be seen at the following link (https://youtu.be/68xWSsWeXBA (accessed on 23 March 2022)).

The result of this experiment was that the network inferred the goal correctly with 60% probability while observing the placing actions with 100% probability while observing the swinging actions. However, we note that the capability of the network to correctly infer the goal and future actions requires the actions of the human grasping the cylindrical object to closely match the robot’s own learned movement image, which is not easy for humans to consistently reproduce. Particularly in the case of grasping-placing, precise timing and position in grasping and placing become critical and the loss of precision in the visual tracking of the object at longer distances posed additional challenges. When the demonstrated actions deviated from those learned by the network, we observed that the free energy increased instead of decreasing over time, which produced unreliable results. A future study should investigate methods of making the network more robust in order to better tolerate unreliable human factors, which would be encountered in real-world settings.

## 5. Discussion

The current study proposed a novel model for goal-directed action, planning, and execution under a teleological framework extending active inference. The proposed model, T-GLean, is characterized by three features. First, goals can be specified either by a specific sensory state expected at a distal step or dynamically changing sensory sequences. Second, goals can be inferred by observed sensory sequences. Third, the goal-directed plan is generated by situating the latent state to the observed sensation by means of the online inference. We have shown that our model is well-suited to accomplishing tasks in continuous sensorimotor space, which is required for embodied cognitive systems such as humanoid robots. At the same time, our model can also manage stochastic noise or fluctuations in real-world systems since the formulation is based on the free energy principle, which deals with Bayesian probability.

The proposed model was evaluated by conducting two experiments, the first using a simulated mobile agent for navigation and the second using a physical humanoid for object manipulation. The results of experiments using a simulated mobile agent showed that generalization in generation of reaching movements to unlearned goal positions is sufficient with a relatively small number of training samples, with modest improvement as the number of teaching trajectories is increased. It was also shown that both types of goal-directed plan generation and their execution, i.e., reaching a specified position and cycling could be performed precisely. Furthermore, it was demonstrated that goals could be inferred adequately from the observed sensation, even in the case of dynamically changing goals. Finally, it was shown that the network could generate goal-directed reaching plans with the shortest path when an additional cost for travel time minimization was added to the original plan free energy formula. This confirms that the current model using this modified plan free energy can generate optimal goal-directed plans under well-posed conditions.

In the results of the experiments scaled up to using a real humanoid robot, it was shown that goal-directed plan generation and execution, as well as goal inference by observation could be performed with reasonable performance for two different goal-directed actions, grasping-placing and grasping-swinging, although their performance was slightly worse compared to the simulated mobile agent case. This could be due to various real-world constraints, including limited precision in the visual tracking system and in motor control, as well as unreliable human behavioral factors in demonstrating emulated goal-directed actions to the robot. There is still plenty of room for improving performance in such real-world situations by making technical efforts in various regards.

Our proposed model is distinct from typical active inference-based approaches in that rather than conducting a policy search by inferring the optimal posterior predictive distribution to minimize the expected free energy, such as in [18,20], we minimize expected free energy in the latent space by gradient descent, and then use this optimized posterior predictive distribution to generate a proprioceptive-exteroceptive trajectory that a controller maps to optimal motor control commands. This allows for more efficient trajectory planning by working in the lower-dimensional probabilistic latent space than in the higher dimensional motor space of robots with many degrees of freedom.

Active inference has also been applied to robot control by directly generating acceleration or torque commands for the robot motors. For instance, an active inference model can perform robust inverse kinematics to accomplish head tracking and reaching behavior [33]. Although the current study focused more on higher level goal-directed planning, in which motor control is conducted by a PID controller outside the proposed model, our future study could incorporate this sort of motor torque scheme. Several works, such as [34], have also used a variational autoencoder to map visual images to a latent space, which is used to generate optimal motor control commands. This approach generates goal-seeking actions even without explicitly learning the environment [35]. Processing pixel-level visual images tends to create a computation bottleneck in real-time robot control at this time however, which is a key reason we did not pursue it for this work.

In the field of reinforcement learning (RL), there has been a large amount of work on goal-conditioned planning. For instance, it was shown that a single network can effectively learn to achieve multiple goals, and can then achieve any particular goal [36]. Through exploration, a model-based RL method can learn not only the world, but also goals that can be accomplished [37]. It was also shown that goal inference may be possible through learning of the goal reward [38]. An interesting extension to this is the embedding of multiple sub-goals in a latent space [23], which also enables accurate anticipation of the goal. Although the aforementioned study is analogous to the current study in the way it uses latent space for planning, our study did not attempt to explicitly encode sub-goals, such as reaching, grasping, and placing. The current study did not focus on reinforcement learning based on self-exploration because the error regression scheme applied to the online approximation of the posterior in the past window again poses a computation bottleneck that does not allow a large amount of self-exploration, especially in real robot experiments. Future study, however, should consider this problem.

An interesting extension of this current work would be to examine how robots can deal with unexpected environmental changes, using the current model. For example, if Torobo fails to grasp the object or drops it, can it recover from the failure by generating a new recovery action plan? It would be interesting to examine how much such an unexpected situation can be recognized by inferring the latent state by means of minimization of the evidence free energy applied to the past window. This may require additional learning of various failure situations so that novel situations can be adequately handled through generalization, while maintaining well-posed solutions for normal situations.

Another direction for future study would be further scaling up in action and goal complexity by introducing a language modality. By using the power of rich linguistic expressions, it is expected that various complex goals can be represented in a compositional manner. It is, however, very likely that it will be quite difficult to learn an adequate amount of language relevant to goal-directed actions with different levels of complexity at once. With regard to this problem, one plausible but challenging approach may be the introduction of developmental pathways in learning. It would be natural to start by learning a set of simple goal representations that could be achieved by some primitive behaviors. When learning proceeds further, more complex goal-directed actions could be learned by means of compositions of the prior-learned action primitives associated with corresponding compositional linguistic expressions. This might lead to acquisition of a more abstract goal representation at the conceptual level.

## Figures and Tables

**Figure 1 entropy-24-00469-f001:**
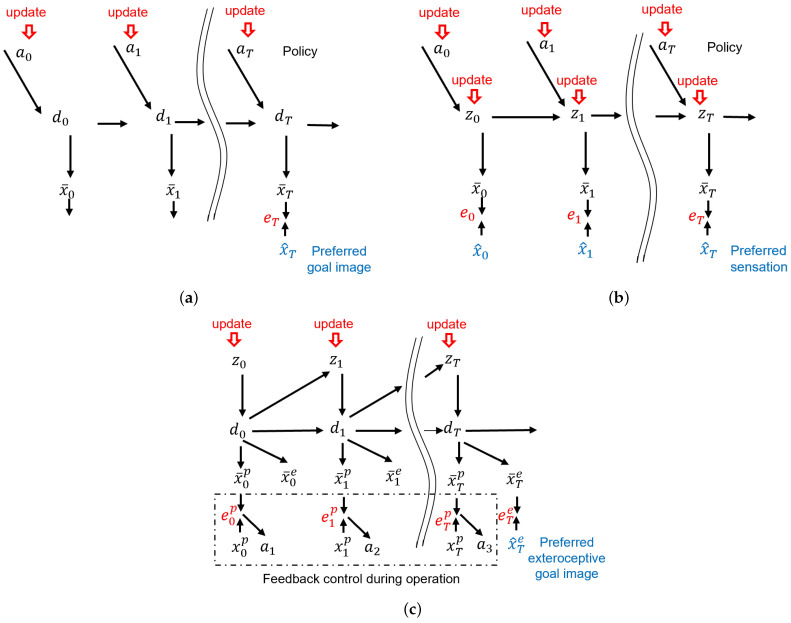
Prior models for generating goal-directed behaviors. (**a**) Forward model using latent variables, (**b**) active inference model using probabilistic latent variables, and (**c**) GLean as an extension of active inference.

**Figure 2 entropy-24-00469-f002:**
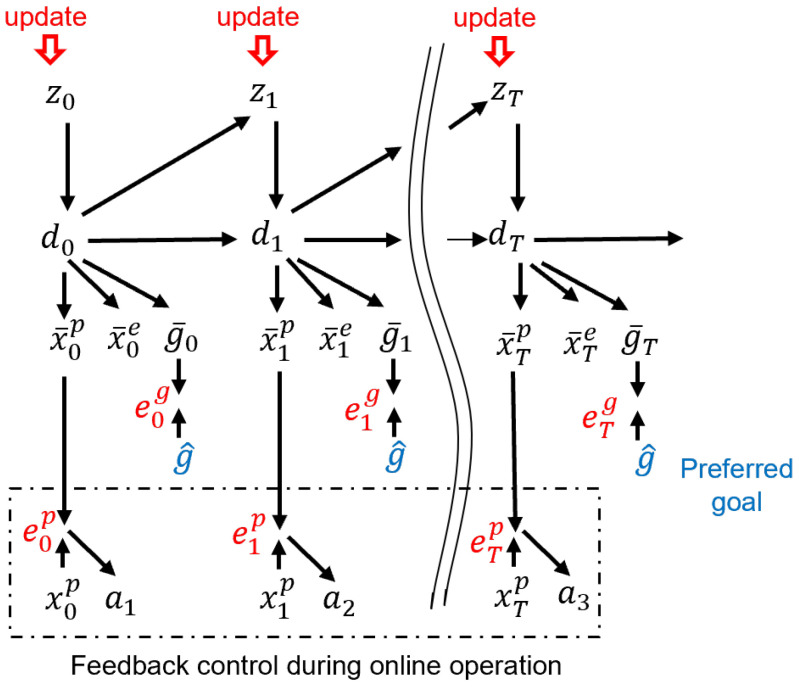
Overview of the newly proposed model.

**Figure 3 entropy-24-00469-f003:**
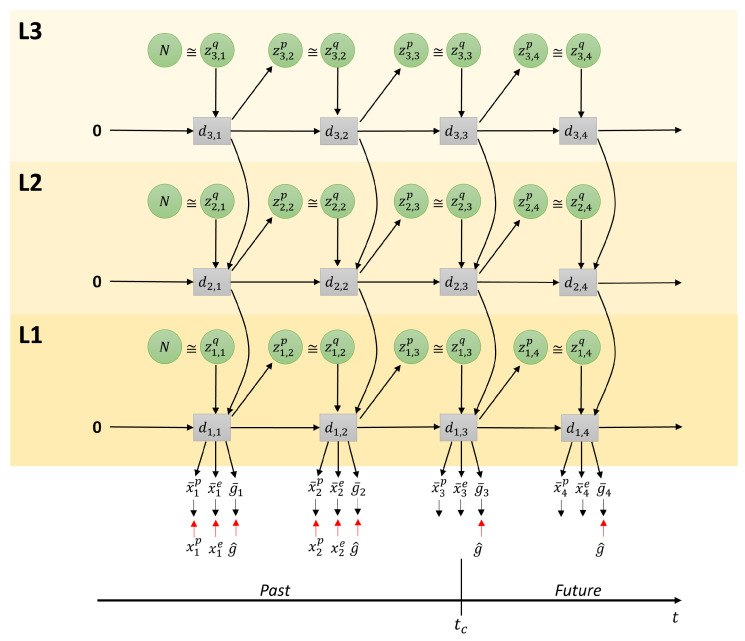
Graphical description of the employed architecture during plan generation.

**Figure 4 entropy-24-00469-f004:**
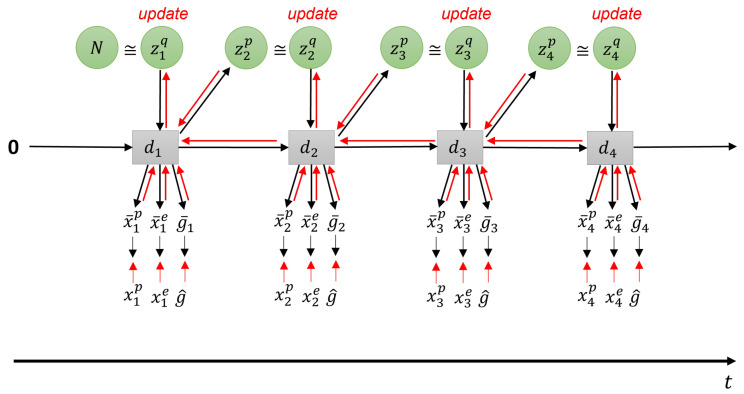
Network during training. Red lines indicate error backpropagation.

**Figure 5 entropy-24-00469-f005:**
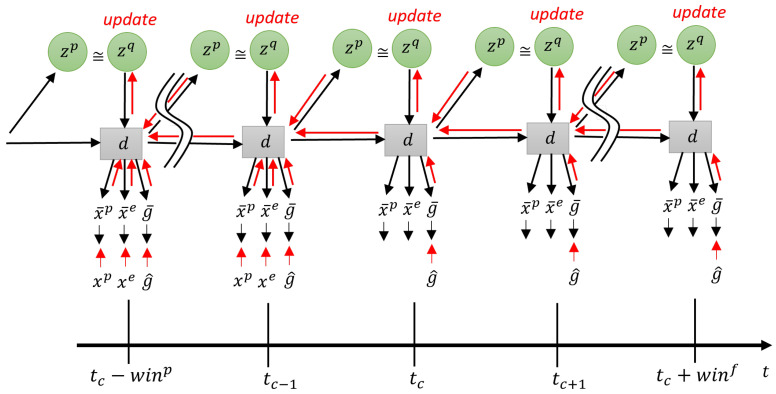
The network during planning with a planning window covering from $t_{c}-win^{p}$ to $t_{c}+win^{f}$ where $t_{c}$ is current time step. Red lines indicate error backpropagation.

**Figure 6 entropy-24-00469-f006:**
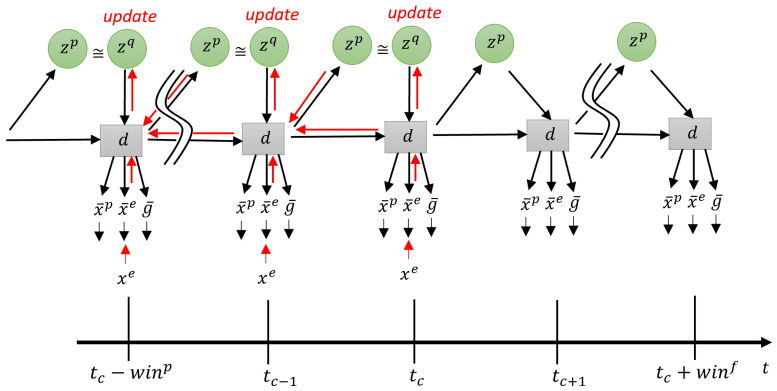
Network during goal inference. Red lines indicate error backpropagation.

**Figure 7 entropy-24-00469-f007:**
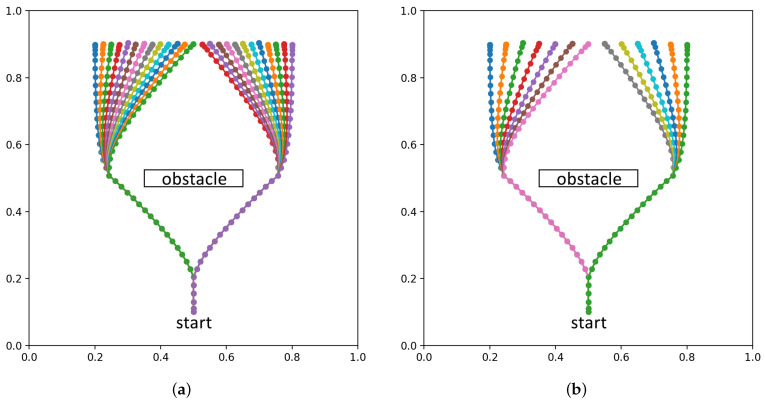
Testing goal position generalization by reducing the number of training samples (**a**) 25 training samples (**b**) 13 training samples (**c**) 7 training samples (**d**) 4 training samples.

**Figure 8 entropy-24-00469-f008:**
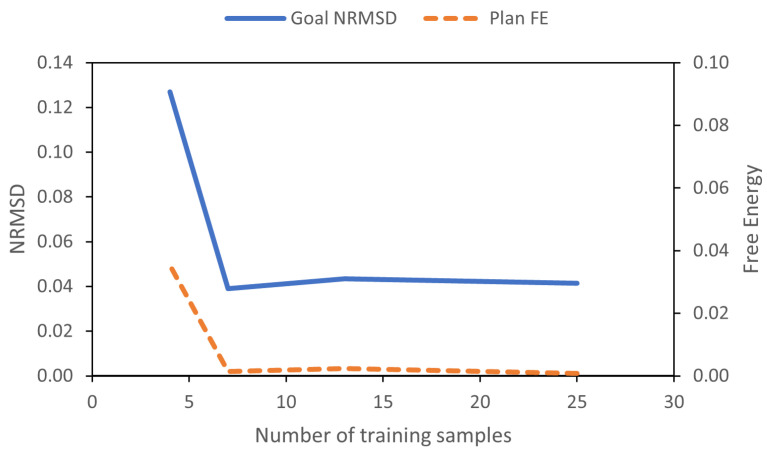
Goal deviation and mean plan free energy (plan FE) for different numbers of training goals.

**Figure 9 entropy-24-00469-f009:**
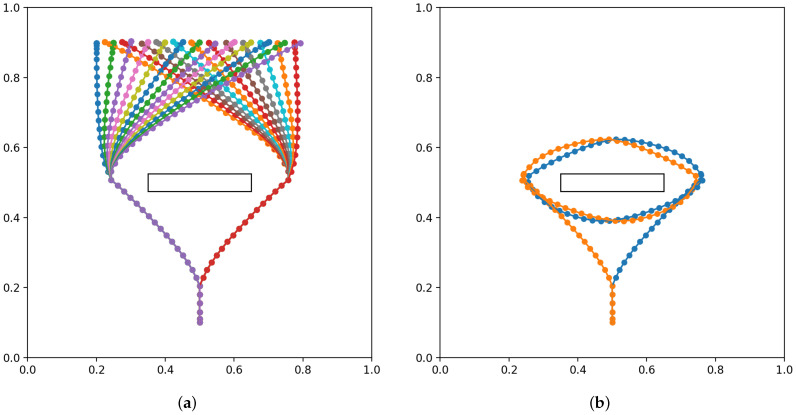
Teaching trajectories used for Experiment 1B, with both goal-reaching and cycling goals. (**a**) 25 reaching trajectories, with both short and long paths (**b**) 2 cycling trajectories, in the clockwise and counter-clockwise directions.

**Figure 10 entropy-24-00469-f010:**
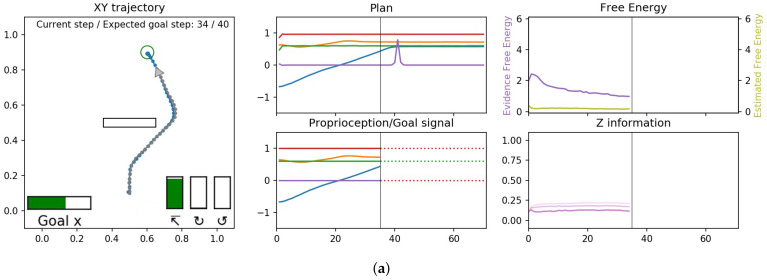
Examples of trajectories generated for three different goal categories, (**a**) reaching, (**b**) clockwise cycle, and (**c**) counter-clockwise cycle from the results of Experiment 1B. The left column shows a view of the 2D workspace, with the generated plan trajectory, shown as a solid line with dots at each time step and the goal position if available. The agent is represented by a triangle, with the agent’s position history trailing behind it and overlaid on the plan trajectory. The subset along the bottom (enlarged for visibility) is the expected goal, the left-most horizontal bar is ${\bar{g}}^{\alpha }$ (goal position), while the three vertical bars is ${\bar{g}}^{\beta }$ (goal category); the left vertical bar represents reaching, the middle bar represents clockwise and the right bar represents counter-clockwise goals. The height of the vertical bar represents the probability (confidence) of this goal. The distal step, if available, is also shown both as a circle in 2D and in text in the top right. The middle column shows the plan in terms of exteroceptive trajectory in the top as well as the observed exteroception trajectory and preferred goal (encoded for display as scalar values 1, 0 and −1 for reaching, clockwise and counter-clockwise, respectively) in dotted lines in the bottom, with the vertical black bar representing the current sensorimotor time step. The right column shows the evidence free energy (dark purple) and the expected free energy (light green) in the top and *z* information in each layer (darker lines are higher layers).

**Figure 11 entropy-24-00469-f011:**
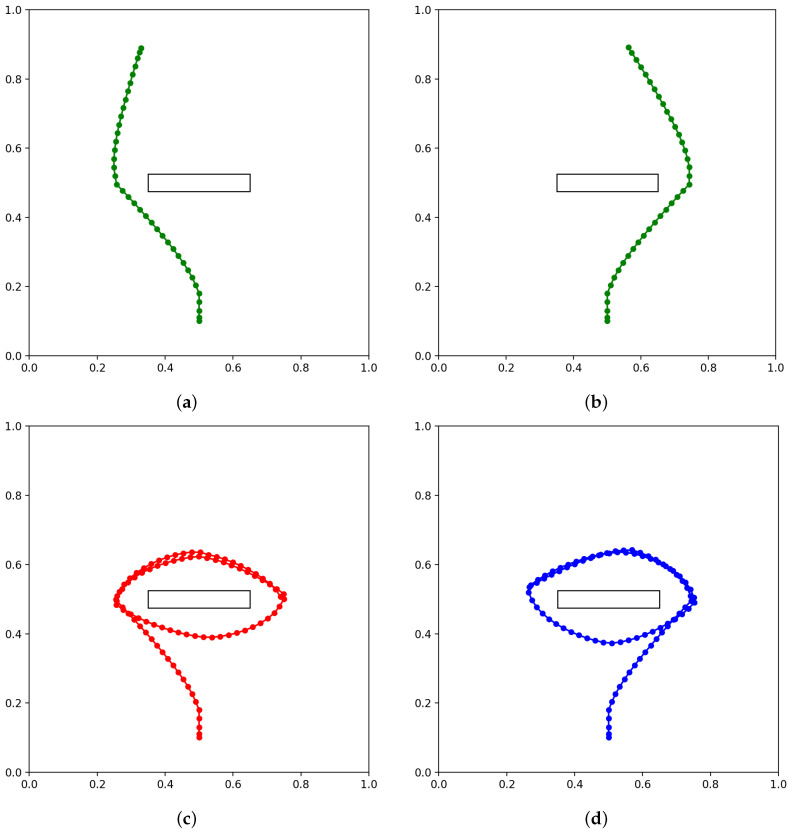
Test trajectories used for goal inference, color coded by the type of goal. These trajectories represent agent actions for (**a**) reaching goal located on the left, (**b**) reaching goal located on the right, (**c**) clockwise cycling, and (**d**) counter-clockwise cycling. The reaching trajectories and the cycling trajectories are 40 and 90 steps long, respectively.

**Figure 12 entropy-24-00469-f012:**
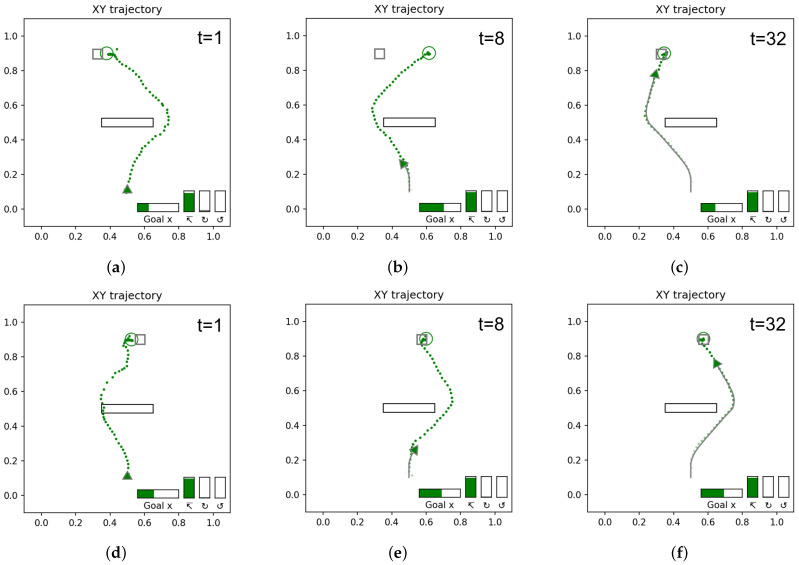
Anticipated goal and trajectories as the network observes the reaching trajectories at different time steps during the observation of travels. The colored circles represent anticipated goals, and gray squares represent actual goals. The colored dots represent the anticipated trajectory, while the solid gray line is the observed trajectory. The time step is shown in the top right. (**a**–**c**) Agent observing the trajectory reaching to the left goal, (**d**–**f**) agent observing the trajectory reaching to the right goal. Note that for both goals, initial anticipated trajectories (**a**,**d**) are significantly different from later (**c**,**f**) trajectories. As the agent moves, the exteroception error in past steps is minimized and this improves the anticipated trajectory. The anticipated goal also becomes closer to the actual goal, as most clearly illustrated in (**b**,**c**).

**Figure 13 entropy-24-00469-f013:**
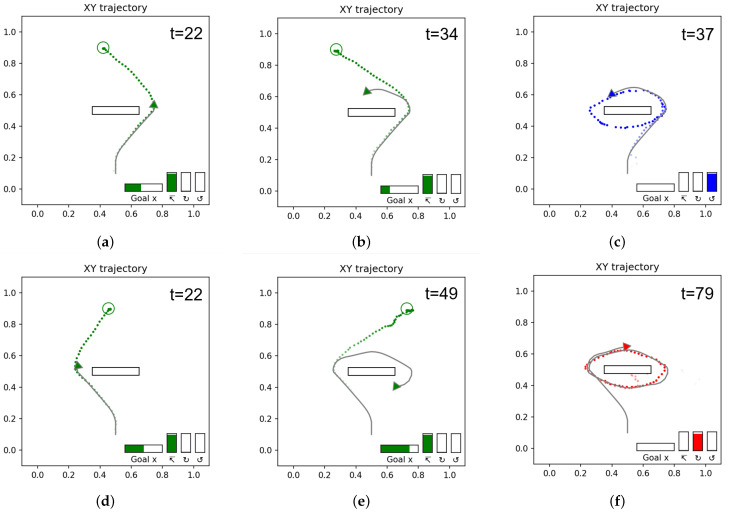
Anticipated goal and trajectories as the network observes the cyclic trajectories. The colored dots represent the anticipated trajectory, while the solid gray line is the observed trajectory. The time step is shown in the top right. (**a**–**c**) Agent following the counter-clockwise cyclic goal trajectory, (**d**–**f**) agent following the clockwise cyclic goal trajectory.

**Figure 14 entropy-24-00469-f014:**
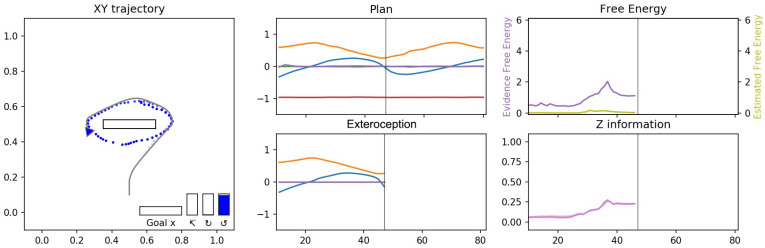
Network inferring the counter-clockwise cycling goal while observing the movement trajectory in the left panel. The center panel shows the inferred plan in the top and the observed exteroception in the bottom. Note the peak in the free energy before the goal is correctly inferred in the right panel.

**Figure 15 entropy-24-00469-f015:**
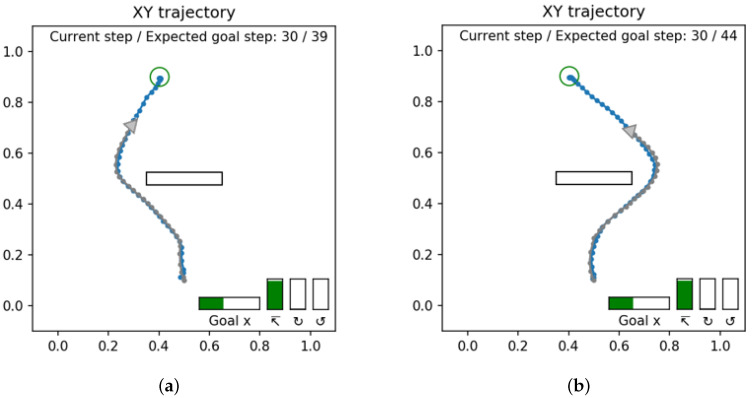
Examples of ill-posed goal-directed planning. (**a**) Generation of a short path reaching the goal (39 time steps) and (**b**) an alternative long path to the same goal (44 time steps).

**Figure 16 entropy-24-00469-f016:**
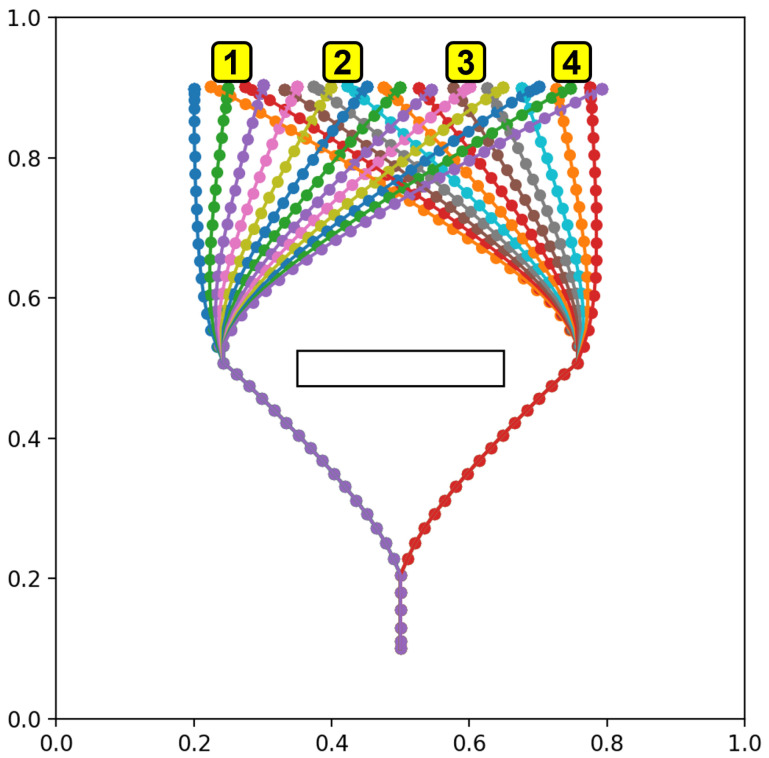
Training trajectories overlaid with the four untrained test goals.

**Figure 17 entropy-24-00469-f017:**
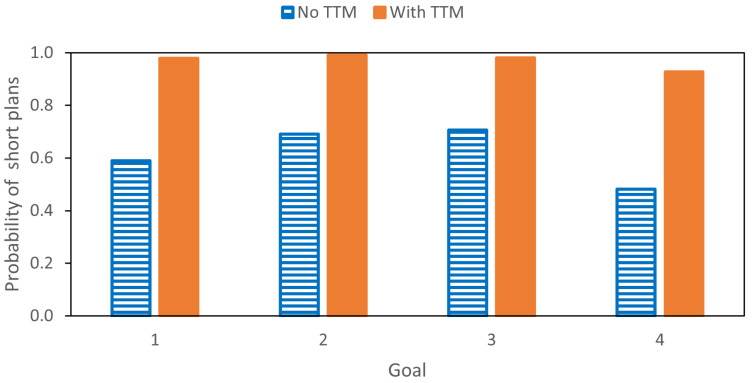
The probability of generating shorter plans for different test goal positions with and without adding the travel time minimization (TTM) constraint.

**Figure 18 entropy-24-00469-f018:**
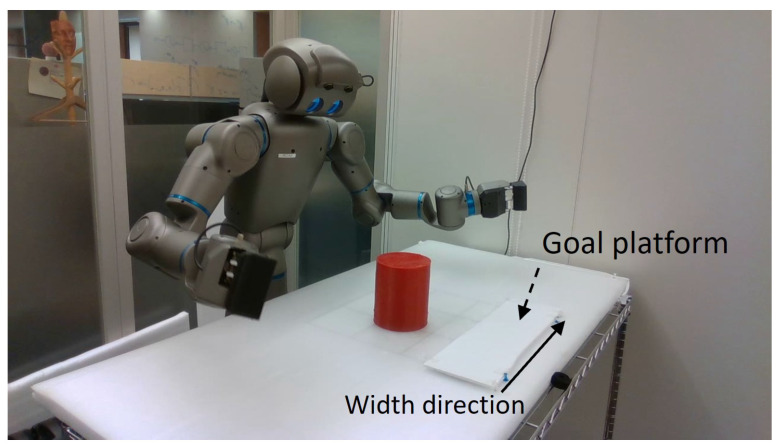
The Torobo humanoid robot, with the workspace, goal platform and object.

**Figure 19 entropy-24-00469-f019:**
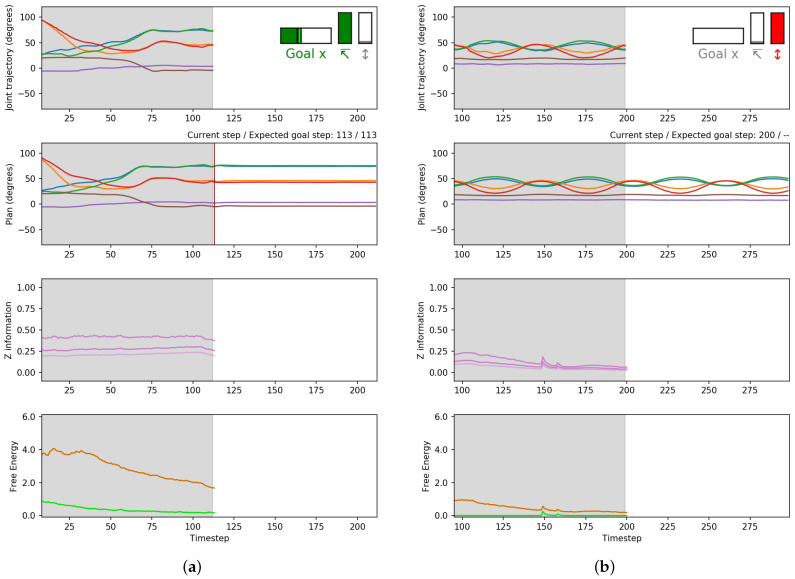
Plots generated while the robot is executing the goal-directed plan. Top to bottom: the sensorimotor history for each robot joint (goal inset on the top right, enlarged for visibility), current planned trajectory for each robot joint, *z* information, and free energy (dark orange: evidence FE, light green: expected FE). Note that only a selected number of joint angles are shown for clarity. (**a**) An example of grasping-placing, as it reaches the distal step (red line), and (**b**) an example of grasping-swinging, which can cycle indefinitely.

**Table 1 entropy-24-00469-t001:** PV-RNN parameters for Experiment 1. $R^{d}$ and $R^{z}$ refer to the number of deterministic (*d*) units and probabilistic (*z*) units, respectively. $w_{t=1}$ refers to the meta-prior setting at the first time step (in our previous work this was referred to as $w_{I}$).

	Layer		
	1	2	3
$R^{d}$	60	40	20
$R^{z}$	6	4	2
τ	2	4	8
*w*	0.0001	0.0005	0.001
$w_{t=1}$	1.0	1.0	1.0

**Table 2 entropy-24-00469-t002:** Deviation from the ground truth, given as normalized RMS.

	Reaching	Cycling
NRMSD	0.033241	0.028158

**Table 3 entropy-24-00469-t003:** PV-RNN parameters for Experiment 2. Parameter settings are identical to Experiment 1, only using a larger τ to compensate for longer sequences.

	Layer		
	1	2	3
$R^{d}$	60	40	20
$R^{z}$	6	4	2
τ	2	10	20
*w*	0.0001	0.0005	0.001
$w_{t=1}$	1.0	1.0	1.0

**Table 4 entropy-24-00469-t004:** Deviation from the ground truth for Experiment 2A, given as normalized RMS.

	Grasping-Placing	Grasping-Swinging
NRMSD	0.10053	0.01514

## Data Availability

Instructions and source code for reproducing the simulated agent results, along with the datasets used, are available at https://github.com/oist-cnru/T-GLean (accessed on 23 March 2022).

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
