# Peer review of "Goal-Directed Planning and Goal Understanding by Extended Active Inference: Evaluation through Simulated and Physical Robot Experiments"

_entropy, 2022, doi:10.3390/e24040469_

Round 1
Reviewer 1 Report
This work introduces goal-directed planning and goal understanding by active inference and evaluates it using simulation and physical experiments. The results are in general interesting and reasonable.
-A question about the simulation task: all the cases consider one obstacle in the middle with the same shape. Is it possible to use the trined model to cross different obstacles? Please comment on this and also how to extend the model to handle different types of obstacles.
-It seems that the training data in Experiment 2 is not sufficiently introduced. Please specify the number of training samples, and what type of motion is used.
Reviewer 2 Report
A pdf is uploaded.

Reviewer 3 Report
In this paper, the authors present an extension of their previous work GLean, called T-GLean. The core idea and contribution is to have the agent form beliefs and infer the goal it is pursuing, modeled within the active inference framework. This has a number of advantages, as it allows for a more general goal definition, i.e. goals can be reprsented by a number of demonstrated trajectories, rather than a single target state or observation, and it enables an agent to infer ones goal from a followed trajectory. The authors clearly demonstrate their approach through a number of experiments, both in simulation as on a real-world robot.
Although I really like the idea, I have two major concerns with the current form of the manuscript.
First, I found section 3 that explains the approach hard to follow, and could benefit from a restructuring. For example, some terms are introduced in text beforehand, but only defined later (i.e. "the evidence free energy"), whereas until that point (line 175) only (two flavours of) the expected free energy was mentioned in section 2. Also, details of how the model is instantiated (i.e. lines 210-216), or how terms are calculated (i.e. lines 225-226) are mixed with more generic explanations (i.e. the objective functions minimized). I think it would be better to first have a high level description of the generative model (Fig.3), followed by a description of free energy, expected free energy, planning and goal inference, and only afterwards discuss the actualy instantiation of the model with the diferent weights and activation functions.
Some aspects can also use some more explanation, such as the distal probabilty s, which was unclear to me after a first read.
Second, the authors mainly discuss their work in the context of their previous work, but I think a more elaborate discussion on state of the art is appropriate. Some suggestions:
On learning recurrent state space models for active inference:
Fountas et al. "Deep active inference agents using Monte-Carlo methods"
Catal et al. "Learning Generative State Space Models for Active Inference"
Sajid et al. "Exploration and preference satisfaction trade-off in reward-free learning"
On active inference for robot control:
Oliver et al. "An empirical study of active inference on a humanoid robot"
Meo et al. "Adaptation through prediction: multisensory active inference torque control"
Van de Maele et al. "Active vision for robot manipulators using the free energy principle"
Also the field of (deep) reinforcement learning has large body of work on goal-conditioned planning/policies which I think is relevant for the discussion:
Andrychowicz et al. "Hindsight Experience Replay"
Nasiriany et al. "Planning with Goal-Conditioned Policies"
Mendonca et al. "Discovering and Achieving Goals via World Models"
Warde-Farley et al. "Unsupervised Control Through Non-Parametric Discriminative Rewards"
Rudner et al. "Outcome-Driven Reinforcement Learning via Variational Inference"
Some minor remarks:
- l106: "by extending the framework of active inference based on the free energy principle" - active inference is a corollary of the FEP, so it's not extended based in this, or am I misinterpreting this sentence?
- Equation 1: extrinsic value uses log P(.), as probability distribution function P versus the density function p used elsewhere.
- l201: "are softmax encoded", I assume the targets are one-hot encoded, and the outputs are fed through a softmax function?
- l225: "complexity, is computed as the KL divergence between the prior and approximate posterior", this should be the other way around, the KL between the approximate posterior and the prior, given that KL is not symmetric.
- In the simulation experiments, is the heading of the agent represented by the triangle, or not?
- In the real-world experiment, how are the training trajectories generated?
- In the real-world experiment, how is the ground truth and final position obtained? Is this calculated from the camera pose and Yolo detector, or measured on the workspace? What is the measurement error on this?
- Some figures are very hard to read, and should be presented bigger (i.e. Figures 1, 2) but also for the results figures, use the complete page width to improve readability of the axes labels.
